# The Real-Life Use of a Protein-Sparing Modified Fast Diet by Nasogastric Tube (ProMoFasT) in Adults with Obesity: An Open-Label Randomized Controlled Trial

**DOI:** 10.3390/nu15224822

**Published:** 2023-11-17

**Authors:** Elena Formisano, Irene Schiavetti, Raffaella Gradaschi, Paolo Gardella, Carlotta Romeo, Livia Pisciotta, Samir Giuseppe Sukkar

**Affiliations:** 1Department of Internal Medicine, University of Genoa, 16132 Genoa, Italypaolo.gardella@yahoo.it (P.G.);; 2Dietetics and Clinical Nutrition Unit, IRCCS Ospedale Policlinico San Martino, 16132 Genoa, Italy; 3Department of Health Sciences, Section of Biostatistics, University of Genova, 16132 Genova, Italy

**Keywords:** protein-sparing modified fast, continuous enteral feeding, weight loss, severe obesity, VLCKD, whey protein

## Abstract

Background: Protein-sparing modified fast (PSMF) diet is a very-low-carbohydrate ketogenic diet administered to patients with obesity, which preserves lean mass and suppresses appetite as well as continuous enteral feeding. Thus, we aim to evaluate the effect of the PSMF diet administered continuously by nasogastric tube (NGT) or orally. Methods: Patients with a body mass index (BMI) > 34.9 kg/m^2^ were randomly assigned to receive a whey protein PSMF formula through NGT (ProMoFasT) or orally. Data were collected at baseline and after 150 days. The endpoints were assessed in the intention-to-treat population. Results: We enrolled 20 patients in the ProMoFasT group and 24 in the oral group. No differences in body weight, BMI or waist circumference between the two groups were found after 150 days. At follow-up, FFM (%) and MM (%) results were higher in the ProMoFasT group than the oral group (63.1% vs. 52.9%, *p* = 0.012 and 45.0% vs. 36.1%, *p* = 0.009, respectively) and FM (kg) and FM (%) were significantly lower in the ProMoFasT group (36.9 kg vs. 44.0 kg, *p* = 0.033 and 37.4% vs. 44.9%, *p* = 0.012, respectively). Insulin levels were lower in the ProMoFasT group than the oral group at follow-up (11.8 mU/L vs. 28.0 mU/L, *p* = 0.001, respectively). Conclusion: The ProMoFasT is more effective in improving body composition and glucometabolic markers than the same diet administered orally.

## 1. Introduction

In recent decades, worldwide obesity has tripled and is currently reaching epidemic proportions, since in 2016 over 650 million people were affected by obesity [1,2,3]. This pathological condition leads to a significant increase in morbidity and mortality, derived from its associated chronic comorbidities, including arterial hypertension, dyslipidemia, type 2 diabetes mellitus and cardiovascular diseases [4,5]. 

Traditional approaches for managing obesity, such as lifestyle modification with a balanced Mediterranean diet, regular exercise, behavior therapy and drug administration, generally suffer from low long-term compliance [6,7]. Bariatric surgery is a valid therapeutic option for patients with severe obesity (BMI ≥ 40 kg/m^2^ alone or BMI ≥ 35 kg/m^2^ with significant comorbidities) that leads to persistent weight loss and could reduce obesity-related comorbid illnesses [8,9]; however, the surgical option is a procedure associated with significant short- and long-term risks [10], and improvements in comorbid conditions vary among the currently available studies, depending also on the procedure, amount of weight loss attained, effect on eating habits, alterations in hormones and incretins, degree of malabsorption and change in motility [11,12,13].

In clinical practice, very-low-carbohydrate ketogenic diets (VLCKDs) are commonly used to achieve rapid weight loss, particularly in patients affected by severe or complicated obesity (i.e., metabolic disorders, obstructive sleep apnea or severe arthropathies) and in patients who need rapid weight loss for severe comorbidities or for scheduled surgery [14,15]; moreover, VLCKDs represent an effective nutritional strategy for subjects that are not eligible for bariatric surgery [16]. 

The protein-sparing modified fast (PSMF) diet is a VLCKD with a high-quality protein content (1.2–1.5 g/kg/day), a carbohydrate amount less than 20–30 g/day and a low-fat intake (about 10–20 g/day); additionally, it requires a significant amount of water (above two liters) as well as vitamin and mineral supplementation [15]. According to Blackburn et al., the PSMF diet exhibits a protein-sparing effect, based on the rationale that the administration of amino acids during fasting without glucose induces a much smaller insulin response compared to the administration in combination with glucose, minimizing the antilipolytic effect of insulin [17]. However, it should be considered that the protein intake is interrupted throughout the night, resulting in protracted overnight fasting with muscle catabolism and a decrease in lipolytic action [18]. 

Thus, the continuous enteral nutrition (EN) of a high-protein diet administered through a nasogastric tube (NGT) may be a valid approach for counteracting the catabolism of lean mass [18]. Moreover, Beale et al. observed a greater suppression of appetite and increased levels of glucagon-like peptide 1 and peptide YY with a single mixed-meal bolus via NGT compared to the same meal taken orally [19]; additionally, NGT feeding could bypass the limited poor dietary adherence characterizing VLCKDs [20,21]. Supporting this evidence, Castaldo et al. highlighted that a weight-loss-based EN strategy is an effective and safe approach for improving body mass index (BMI), waist circumference, as well as glycemic and lipid profiles [21]. Similarly, in a trial conducted by Sukkar et al., twenty-two patients with obesity experienced a decrease in total body weight and waist circumference, as well as an improvement in respiratory capacity after 10 days on an enteral PSMF diet followed by 20 days on a low-calorie diet, without side effects [22]. 

To date, no evidence reports a comparison between a PSMF diet administered continuously via NGT and the equivalent PSMF diet taken orally on anthropometric and body composition data in patients with obesity. Therefore, our study aims to evaluate the effect of a PSMF diet on anthropometric and body composition parameters when supplied continuously by NGT or administered per os. Secondly, we examine the impact of the PSMF diet on blood glucometabolic markers.

## 2. Materials and Methods

### 2.1. Study Design and Subjects

The present study is a pilot single-center randomized clinical trial designed to test the efficacy of a PSMF diet administered via NGT (ProMoFasT) compared with the same PSMF diet taken orally in patients with obesity. Subjects were evaluated and enrolled in the Dietetics and Clinical Nutrition Unit of the IRCCS Policlinic San Martino Hospital, University of Genoa (Italy) according to the following inclusion and exclusion criteria. 

We included patients with obesity with a BMI above 34.9 kg/m^2^ whose age ranged from 18 to 70 years who had not previously responded to weight-lowering drugs and/or cognitive/behavioral therapy. Exclusion criteria were Type 1 Diabetes Mellitus, Types 2 Diabetes Mellitus (T2DM) treated with insulin, pregnancy and breastfeeding, kidney failure and severe chronic kidney disease, liver failure, multi-organ failure, intestinal bowel diseases, eating disorders and other severe mental illnesses, alcohol and drug abuse, active neoplasm, malignant hematological disease, severe esophagitis, voluminous symptomatic hiatal hernia (hiatal surface area > 5 cm), active stage of gastric and/or duodenal ulcers, previous restrictive surgery of the gastrointestinal tract, esophageal-gastro-duodenal bleeding or potential bleeding, patients treated with nonsteroidal anti-inflammatory drugs and anticoagulants and patients with contraindication to EN.

Informed written consent for the use of personal data was obtained from patients. The study was conducted in accordance with the Declaration of Helsinki and was approved by the Ethics Committee of the IRCCS Policlinic Hospital San Martino in Genoa (Italy) (n.reg CEA 125/10) in February 2011 and was registered on clinicaltrials.gov (protocol record: EMF2011, id.n.NCT01538654).

### 2.2. Nutritional Protocol

Patients were randomized 1:1 to a PSMF diet administered continuously during 24 h via NGT (ProMoFasT) or to the same PSMF diet administered orally. The protocol included 5 cycles of active PSMF nutrition lasting 10 days alternated to 20 days of a balanced oral low-calorie diet for a total of 150 days. The PSMF diet administered in both two groups was based on a whey protein formula (Table 1), a multivitamin/mineral supplement and a fiber supplement. The balanced oral low-calorie diet presented a caloric deficit of about 10% of the patient’s caloric needs assessed by indirect calorimetry (SenseWear Pro3 Armband; BodyMedia, Pittsburgh, PA, USA). Body weight, height and waist circumference were recorded, and body mass index (BMI) was assessed at baseline and at a follow-up visit after 150 days. Body composition was determined by bioelectrical impedance analysis (BIA) AKERN BIA 101 (BIA 2000-S; Akern srl, Firenze, Italy) and the results were analyzed using dedicated software (Bodygram Plus^®^ v.1.0, Akern 2014, Firenze, Italy). BIA analysis was performed at baseline and at the 150-day follow-up visit and included the following parameters: standardized phase angle (PA) (°), body cell mass (BCM) (kg), fat-free mass (FFM) (kg and %), muscle mass (MM) (kg and %), fat mass (FM) (kg and %), total body water (TBW) (L and %) and intracellular water (ICW) (L and %). Blood test analysis was assessed and the following data were recorded at baseline and at the follow-up visit: fasting blood glucose, fasting blood insulin, glycated hemoglobin (HbA1c), homeostasis model assessment of insulin resistance (HOMA-IR), creatinine, gamma-glutamyl transferase (GGT), alkaline phosphatase (ALP), aspartate amino transferase (AST), alanine amino transferase (ALT), total cholesterol (TC), high-density lipoprotein cholesterol (HDL-C), low-density lipoprotein cholesterol (LDL-C), triglycerides (TG), potassium, calcium and phosphorus.

The endpoints were assessed in the intention-to-treat population. 

### 2.3. Naso-Gastric Tube Placement and Home’s Enteral Nutrition Management

After a 12-h overnight fast, an 8 French polyurethane NGT was placed in a day-hospital setting in the Dietetics and Clinical Nutrition Unit of the IRCCS Policlinic San Martino Hospital, University of Genoa, Italy. Patients were provided with tools to carry out home EN and they were elucidated about the use of the infusion pump and any possible side effects of EN.

### 2.4. Statistical Analysis

IBM SPSS Statistics, Version 25.0 (SPSS Inc., Chicago, IL, USA, www.spss.com, accessed on 13 November 2023) was used for statistical analysis. The normality of variables was examined using a Kolmogorov–Smirnov analysis. Contingency tables and ordinal and nominal variables were used to represent frequency and percentage in the population. The median and interquartile range were used to express the results of continuous variables. Data were analyzed according to an intention-to-treat (ITT) principle, which included all 44 randomized subjects. Non-parametric tests (Kruskal–Wallis or Mann–Whitney) were used when necessary to compare continuous variables between different patient groups. For the correlation with continuous variables, the Pearson chi square (X^2^) test and Spearman’s rank correlation index were used to analyze nominal variables. Within-group differences from baseline to follow-up were analyzed by the Wilcoxon’s signed-rank test.

## 3. Results

### 3.1. Study Population

Twenty patients (75.0% females and 25.0% males; median age 49.0 years, IQR 41.0–53.0) were randomized to the administration via NGT (ProMoFasT group), while 24 patients (58.3% females and 41.7% males; mean age 49.0 years, IQR 39.0–56.5) were assigned to the PSMF diet administration orally (oral group) (Figure 1). Twelve patients, six from each intervention group, discontinued the diet because of intolerance at a median of 60 days. The general characteristics of the two groups at baseline are shown in Table 2.

### 3.2. Anthropometric Parameters and Body Composition Analysis

Table 3 shows that body weight, BMI and waist circumference were significantly reduced in both groups from baseline to follow-up visit, although no significant differences between the two groups were found among all these parameters after 150 days of the PSMF diet. Figure 2 shows the comparison of the body weight between the two groups. At follow-up visit, FFM (%) and MM (%) results were significantly higher in the ProMoFasT group than the oral group (63.1%, IQR 57.6–68.0 vs. 52.9, IQR 48.5–59.0, *p* = 0.012 and 45.0%, IQR 37.4–49.0 vs. 36.1, IQR 33.3–41.0, *p* = 0.009, respectively) (Figure 3) and both FM (kg) and FM (%) were significantly lower in the ProMoFasT group compared to the oral group (36.9 kg, IQR 29.9–41.3 vs. 44.0, IQR 39.6–59.2, *p* = 0.033 and 37.4%, IQR 32.3–43.0 vs. 44.9, IQR 41.4–52.0, *p* = 0.012 respectively) (Figure 4). TBW (%) and ICW (L) were significantly higher in the ProMoFasT group than the oral group after 150 days of PSMF (46.2%, IQR 42.2–49.9% vs. 38.7%, IQR 35.4–43.4%, *p* = 0.012 and 25.0, IQR 20.6–30.2 vs. 21.3, IQR 10.0–23.2, *p* = 0.020, respectively).

### 3.3. Biochemical Parameters

As shown in Table 4, insulin levels were significantly lower, while HbA1c levels results were significantly higher in the ProMoFasT group compared to the oral group after 150 days of PSMF (11.8, IQR 8.0–18.0 vs. 28.0, IQR 22.0–30.2, *p* = 0.001 and 6.2, IQR 5.7–6.4 vs. 5.4, IQR 5.3–5.7, *p* = 0.009, respectively). The fasting blood glucose and HOMA-IR were significantly reduced in both groups from baseline to follow-up visit, although no significant differences between the two groups were found among the latter parameters after 150 days. Figure 5 shows the comparison of the fasting blood insulin and HOMA-IR between the two groups.

TC, LDL-C and TG plasma levels were significantly reduced, and HDL-C plasma levels were significantly increased in the ProMoFasT group from baseline to follow-up visit, while TC, HDL-C and LDL-C levels were significantly increased, and TG levels were significantly reduced in the oral group from baseline to follow-up visit. Nonetheless, no significant differences in the lipid profile were observed between the two groups after 150 days. Calcium and phosphorous levels were significantly increased in both groups from baseline to follow-up visit.

## 4. Discussion

VLCKDs are valid and well-recognized strategies in the treatment of severe obesity and the PSMF diet falls into this group of dietary interventions, but its peculiarity is that not only carbohydrates are restricted to less than 30 g/day, but also lipids are limited to less than 20 g/day, making proteins the primary source of calories. This dietary approach seems to have a protein-sparing effect, preventing muscle catabolism. Furthermore, the consistent administration of a whey-protein-rich diet is more effective in preventing lean mass catabolism and inducing appetite suppression [23,24,25]. This is likely achieved through the stimulation of the Protein kinase B (AKT) pathway, facilitated by a rapidly absorbed whey protein formula with a high protein synthesis rate, distinguishing it from other protein sources [18,22,26,27,28].

Given this knowledge, we found that the PSMF diet administered by NGT is an effective and safe dietary intervention with a high level of compliance that leads to a rapid improvement in body composition parameters and glucometabolic markers when compared to administering the same diet orally. Specifically, we observed improvements in body composition parameters, with significantly lower levels of FM (kg) and FM (%) and higher levels of FFM (%), MM (%). From a pathophysiological perspective, it is known that nutrition administered via NGT determines a greater suppression of appetite in human subjects due to alterations in appetite-suppressing gut hormones, and may increase the levels of glucagon-like peptide 1 and peptide YY as reported by a recent randomized controlled trial (RCT) [19]. Additionally, a consistent suppression of ghrelin levels is reported with enteral nutrition feeding [29]. Therefore, it is possible to hypothesize that the indirect action of continuous enteral administration of a PSMF diet by NGT could have an anabolic action due to the inhibition of protein catabolism and lipid anabolism in a condition of constant satiety. Consequently, this approach could favor muscle development and the simultaneous inhibition of new adipose tissue deposition. Furthermore, Marsset-Baglieri et al. have shown that in the complete absence of carbohydrates, the diminished insulin response shifts the balance towards lipid oxidation, inhibiting triglyceride synthesis and activating triglyceride lipase through the influence of glucagon [30]. 

This latter remark could be supported by the conflicting results of the improvement in muscle anabolism when using intermittent boluses of amino acids versus their continuous administration [31]. It is possible that the protein synthesis could be stimulated by the continuous rather than intermittent administration of amino acids [32,33], as it has been demonstrated that the synthesis of new proteins is inhibited and returned to baseline rates after 90 and 210 min after protein ingestion in an intermittent administration of amino acids [34]. In addition, the inhibition of the AMPK pathway due to the constant flow of amino acids guaranteed by continuous enteral nutrition may play a pivotal role in the anabolic process contributing to the protein synthesis [35,36]. In summary, our findings support the possibility of preserving lean mass in people with obesity during VLCKD such as a PSMF diet.

Another intriguing finding from our study was the possibility of maintaining lean body mass throughout the administration of whey protein as a sole source of protein in PSMF. Whey proteins are rich in leucine, a branched chain amino acid that stimulates human muscle protein synthesis by activating the mTOR signaling pathway independently by Insulin-like growth factor receptor 1, resulting in a muscle anabolism enhancement. Moreover, whey proteins inhibit food intake and tend to improve oxidative balance in rats with obesity as this source of proteins was associated with a slight increase in total glutathione both in the liver and in the blood as compared to a casein-based diet [24,37].

In our study, we observed a lower level of fasting blood insulin after 150 days of the PSMF diet in the ProMoFasT group compared to the oral group. This effect has several beneficial effects on human health, as it is well known in the literature that an insulin-sensitizing diet can improve strong outcomes in both diabetic and non-diabetic patients such as the reduction in cardiovascular mortality [38]. The difference in fasting blood insulin levels that we found between the two groups could be linked to the greater reduction in peripheral free fatty acid over-secretion—as a results of FM reduction—in the ProMoFasT group through a negative stimulation of the hormone-sensitive lipase, whose overexpression is mediated by hyperinsulinism [39]. Furthermore, the low carbohydrate diet content results in a reduction in insulin secretion, which increases lipolysis in the liver [40]. Nevertheless, we found higher levels of HbA1c after 150 days of PSMF diet in the ProMoFasT group compared to the oral group. This is noteworthy considering the reduction between the baseline and follow-up in both groups in HbA1c and fasting blood glucose. It is know that HbA1c reflects the blood glucose concentration over a three-month period [41]. It is possible that continuous enteral nutrition stimulates gluconeogenesis through an insulin-independent pathway, leading to a continuous increase in blood glucose levels during the PSMF diet [42].

Although it has been established that prolonged exposure to excessive amino acids diminished AMPK activity leading to insulin resistance [43], we observed a significant reduction in HOMA-IR in both groups after 150 days. In fact, as reported by Gleason et al., AMPK seems not to be a negative regulator of nutrient-stimulated insulin secretion in pancreatic β-cells [35]. To note, a trend of greater HOMA-IR reduction has been found in the ProMoFasT group, suggesting a positive impact on metabolism, promoting better glucose homeostasis and insulin sensitivity.

Another interesting result emerging from our study is the significant improvement in the lipid profile of the ProMoFasT group from baseline to follow-up, characterized by a decrease in LDL-C and TG and an increase in HDL-C. On the other hand, patients who received the oral PSMF diet experienced a worsening of LDL-C levels. The different results from baseline to follow-up observed in the ProMoFasT group could be attributed to the levels of fasting blood insulin, which appear to be significantly lower in the ProMoFasT group compared to the oral group. In fact, insulin has a well-recognized stimulatory influence on endogenous cholesterol synthesis [44]. Nevertheless, no significant differences among lipid parameters were observed between the two groups at follow-up comparison. 

Based on our findings, calcium and phosphorus plasma levels increase in both groups from baseline to follow-up, although it is still possible to hypothesize that a protein-sparing diet may not negatively impact bone metabolism. In fact, it is well-known that a higher protein diet raises insulin-like growth factor-1, a crucial mediator of bone health [45], even though this outcome may be influenced by the vitamin/mineral supplementation administered to the study participants. 

Finally, we observed a significant reduction in body weight, BMI and waist circumference during 150-day cycles of PSMF, whether administered via NGT or orally. In this field, a recent study suggested the effectiveness and safety of the PSMF diet as an outpatient weight loss strategy for adolescents affected by severe obesity [26]. Moreover, Sukkar et al. highlighted that the PSMF diet is a VLCKD recommended for patients with severe obesity who have indications for bariatric surgery (i.e., in the pre-operative period) and for patients affected by obesity needing rapid weight loss due to serious comorbidities (i.e., obstructive sleep apnea, severe arthropathies) [15]. Our findings were in line with a recent study conducted by Cincione et al. that highlighted a significant reduction in body weight and BMI after 21 days (−8%) of a revised protein-sparing diet in patients with obesity and T2DM [27]. 

To the best of our knowledge, this is the first RCT that compared the safety and the efficacy in terms of body composition and glucometabolic improvement of a PSMF diet administered by NGT, compared to the same diet supplied orally. The limitations of our study include a small sample size and a relative high number of patients lost to follow-up; thus, the findings should be considered as preliminary. Furthermore, the lack of quality-of-life data could be another limitation of the study; in fact, quality of life encompasses significant aspects of well-being such as physical, mental, and emotional health, which are important for assessing the overall effectiveness and sustainability of any dietary intervention in a real-world setting. Another limitation was the non-statistically different distribution of male gender between the two groups, slightly favoring the ProMoFasT group and which may have an impact on the body composition. Nevertheless, we did not observe any differences among all parameters evaluated with the BIA analysis. 

## 5. Conclusions

Compared to the same PSMF diet given orally, the ProMoFasT is an effective and safe nutritional intervention that has a good compliance rate and results in improvements in body composition with an insulin-lowering effect among subjects with severe obesity. Further RCTs are needed to confirm these preliminary findings.

## Figures and Tables

**Figure 1 nutrients-15-04822-f001:**
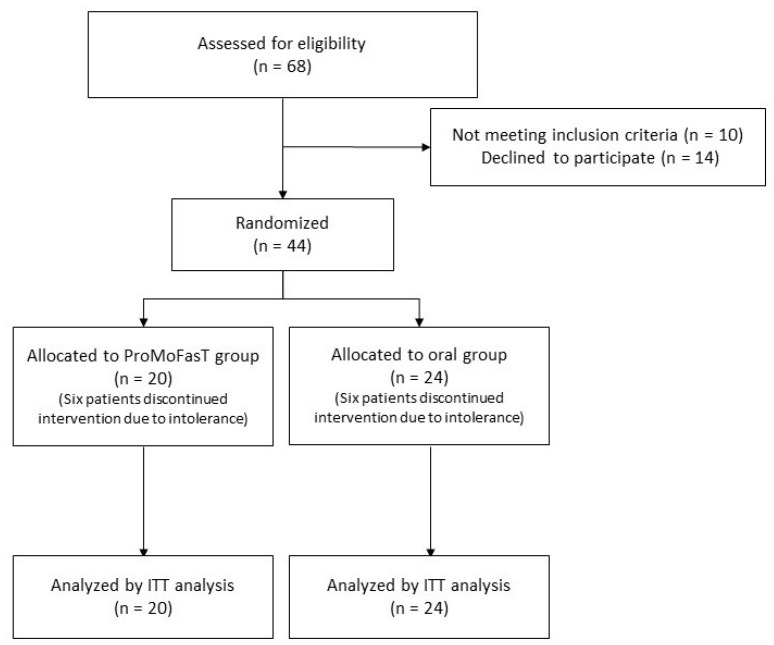
The CONSORT flowchart of the study. Abbreviations: ITT: intention-to-treat; ProMoFasT: Protein-Sparing Modified Fast diet by nasogastric tube.

**Figure 2 nutrients-15-04822-f002:**
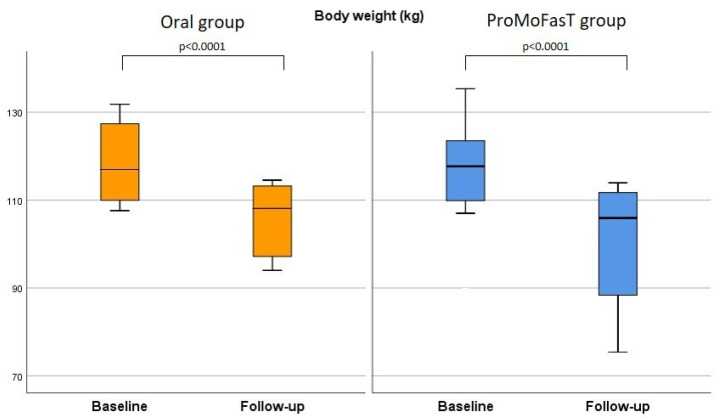
Comparison of the body weight between the two groups. Abbreviations: ProMoFasT: Protein-Sparing Modified Fast diet by nasogastric tube.

**Figure 3 nutrients-15-04822-f003:**
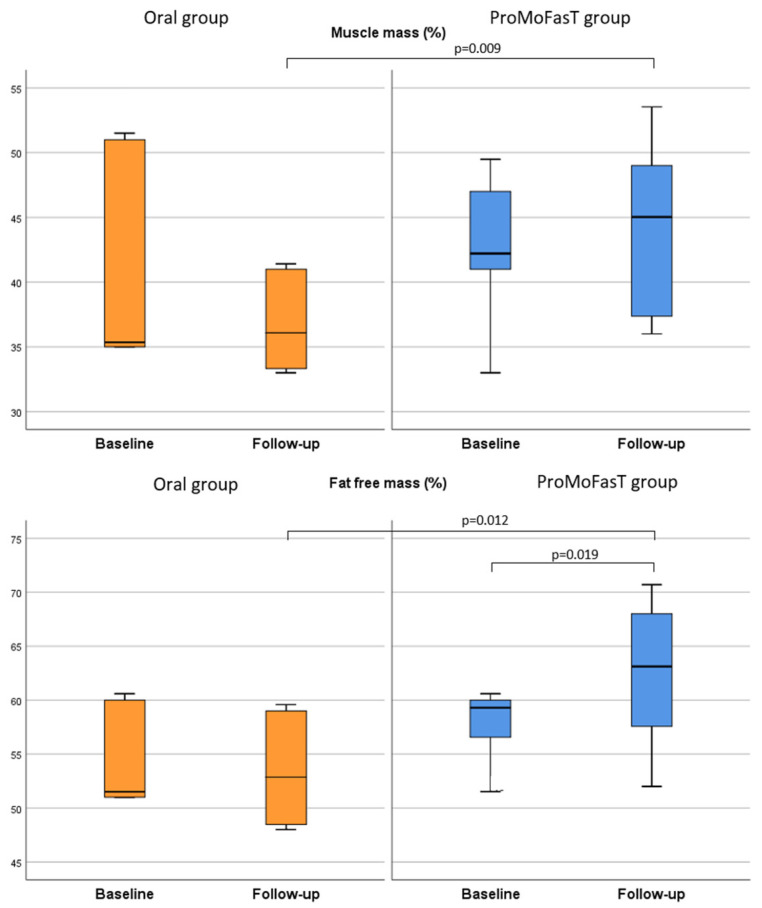
Comparison of the muscle mass (%) and the fat-free mass (%) between the two groups. Abbreviations: ProMoFasT: Protein-Sparing Modified Fast diet by nasogastric tube.

**Figure 4 nutrients-15-04822-f004:**
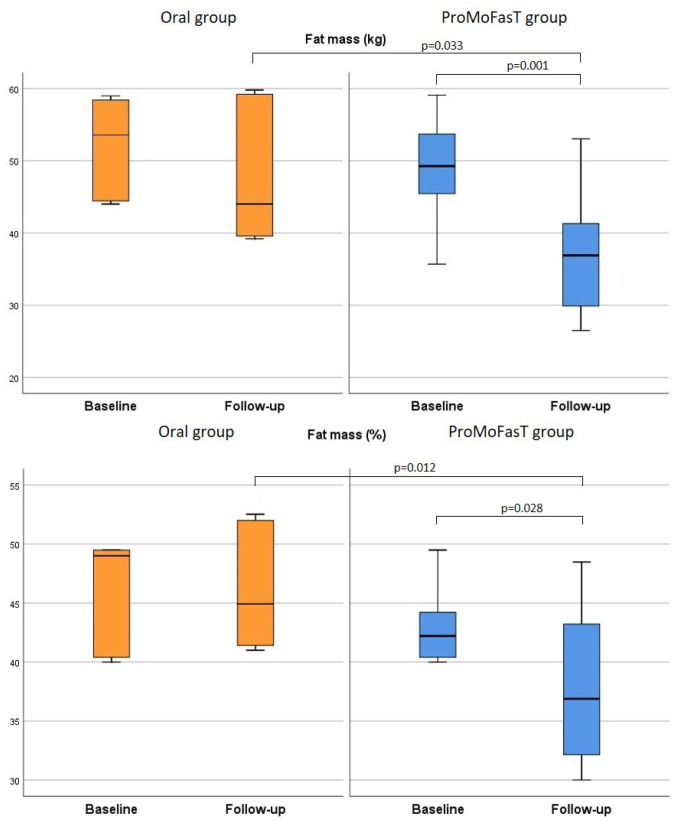
Comparison of the fat mass (kg) and the fat mass (%) between the two groups. Abbreviations: ProMoFasT: Protein-Sparing Modified Fast diet by nasogastric tube.

**Figure 5 nutrients-15-04822-f005:**
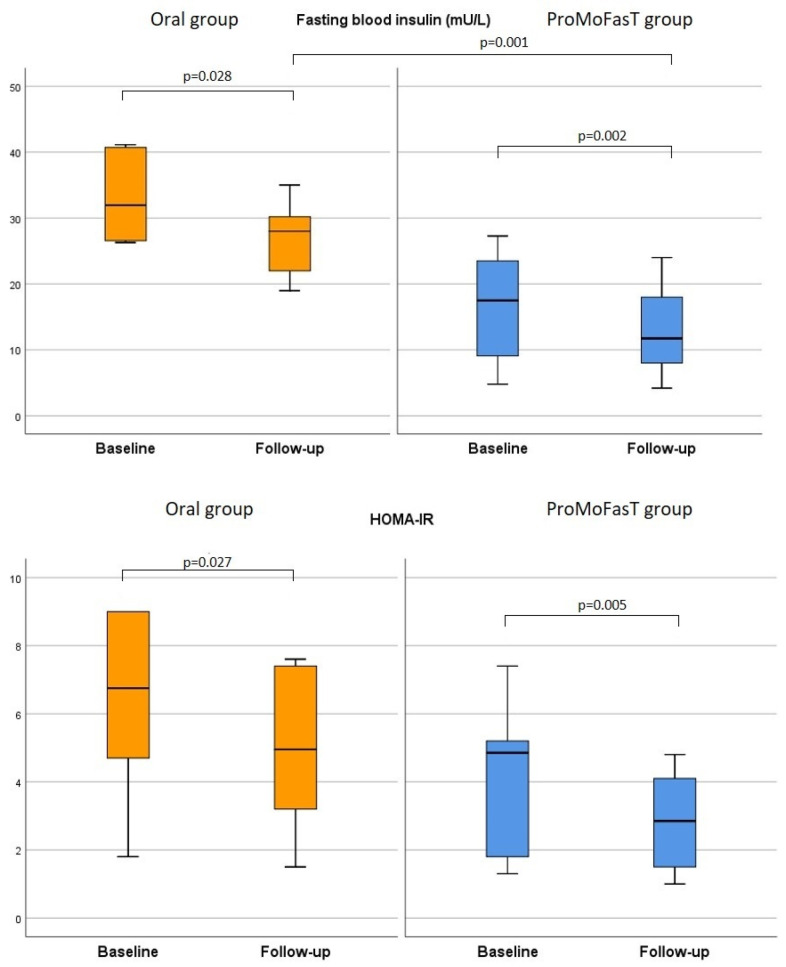
Comparison of the fasting blood insulin and HOMA-IR between the two groups. Abbreviations: HOMA-IR: homeostasis model assessment of insulin resistance; ProMoFasT: Protein-Sparing Modified Fast diet by nasogastric tube.

**Table 1 nutrients-15-04822-t001:** Composition of the whey protein formula administered to both groups.

Nutrients	100 g
Proteins [g]	88
Lipids [g]	1
Carbohydrates [g]	0.3
Fiber [g]	0
Sodium [mg]	550
Potassium [mg]	1170
Calcium [mg]	55
Phosphorus [mg]	220

**Table 2 nutrients-15-04822-t002:** Baseline characteristics of the study population.

	Oral Group (*n* = 24)	ProMoFasT Group (*n* = 20)	*p*-Value
Female gender [*n*, %]	14 (58.3)	15 (75.0)	0.246
Age (years: median; IQR)	49.0 (39.0–56.5)	49.0 (41.0–53.0)	1.000

Abbreviations: IQR: interquartile range; ProMoFasT: Protein-Sparing Modified Fast diet by nasogastric tube.

**Table 3 nutrients-15-04822-t003:** Anthropometrical and body composition parameters between baseline and follow-up and comparison between the two groups after 150 days.

Parameters	Group	Baseline	Follow-Up (150 Days)	*p*-Value (Baseline vs. Follow-Up)	*p*-Value (Oral vs. ProMoFasT)
Body weight (kg)	Oral	117.5 (109.9–127.9)	108.1 (97.2–113.2)	**<0.0001**	0.210
	ProMoFasT	113.9 (102.2–121.6)	105.9 (88.4–111.7)	**<0.0001**	
BMI (kg/m^2^)	Oral	40.8 (40.0–49.4)	37.8 (35.7–44.8)	**<0.0001**	0.564
	ProMoFasT	43.0 (38.3–47.5)	39.3 (32.4–42.9)	**<0.0001**	
Waist circumference (cm)	Oral	129.6 (121.0–132.3)	111.0 (107.5–116.5)	**0.002**	0.527
	ProMoFasT	131.7 (117.0–141.9)	112.5 (103.0–130.0)	**0.001**	
Phase angle (°)	Oral	6.8 (6.4–7.4)	6.5 (6.5–6.6)	0.463	0.153
	ProMoFasT	7.0 (6.4–7.8)	7.1 (6.3–7.5)	0.308	
BCM (kg)	Oral	39.1 (33.1–52.0)	30.5 (27.5–32.4)	**0.028**	0.130
	ProMoFasT	36.3 (31.8–46.0)	34.0 (27.8–41.1)	0.096	
BCM (%)	Oral	57.8 (55.3–60.3)	56.5 (56.0–57.0)	0.465	0.207
	ProMoFasT	58.3 (55.0–61.6)	59.1 (55.6–60.6)	0.060	
FFM (kg)	Oral	67.1 (59.7–86.4)	54.5 (49.1–57.1)	**0.028**	0.207
	ProMoFasT	60.7 (56.8–67.1)	57.4 (53.4–69.9)	**0.035**	
FFM (%)	Oral	56.8 (51.8–64.8)	52.9 (48.5–59.0)	0.463	**0.012**
	ProMoFasT	59.0 (55.6–60.6)	63.1 (57.6–68.0	**0.019**	
MM (kg)	Oral	47.5 (40.8–60.7)	37.2 (33.5–39.5)	**0.028**	0.130
	ProMoFasT	43.8 (38.9–54.7)	41.2 (34.3–49.8)	0.064	
MM (%)	Oral	37.7 (35.7–49.7)	36.1 (33.3–41.0)	0.116	**0.009**
	ProMoFasT	41.4 (38.4–43.4)	45.0 (37.4–49.0)	0.272	
FM (kg)	Oral	50.4 (44.2–56.7)	44.0 (39.6–59.2)	0.116	**0.033**
	ProMoFasT	47.6 (39.2–50.7)	36.9 (29.9–41.3)	**0.001**	
FM (%)	Oral	43.7 (35.7–48.7)	44.9 (41.4–52.0)	0.917	**0.012**
	ProMoFasT	41.2 (40.0–44.4)	37.4 (32.3–43.0)	**0.028**	
TBW (L)	Oral	49.2 (46.1–63.3)	39.9 (35.9–41.8)	**0.028**	0.207
	ProMoFasT	44.8 (41.5–49.3)	42.0 (39.1–51.2)	0.060	
TBW (%)	Oral	42.9 (38.6–47.2)	38.7 (35.4–43.4)	0.463	**0.012**
	ProMoFasT	43.0 (40.4–44.1)	46.2 (42.2–49.9)	**0.028**	
ICW (L)	Oral	29.7 (25.6–38.0)	21.3 (10.0–23.2)	**0.012**	**0.020**
	ProMoFasT	26.5 (23.4–32.9)	25.0 (20.6–30.2)	0.331	
ICW (%)	Oral	58.8 (56.5–60.9)	56.8 (56.3–56.9)	0.463	0.179
	ProMoFasT	58.6 (55.6–61.5)	59.2 (55.8–60.2)	0.826	

Data are expressed as median and interquartile range. No significant differences were found between the two groups at baseline. Abbreviations: ProMoFasT: Protein-Sparing Modified Fast diet by nasogastric tube; BMI: body mass index; BCM: body cellular mass; FFM: fat-free mass; MM: muscle mass; FM: fat mass; TBW: total body water; ICW: intracellular water. Bold font indicates statistical significance.

**Table 4 nutrients-15-04822-t004:** Blood parameters between baseline and follow-up and comparison between the two groups after 150 days.

Parameters	Group	Baseline	Follow-Up (150 Days)	*p*-Value (Baseline vs. Follow-Up)	*p*-Value (Oral vs. ProMoFasT)
Fasting blood glucose	Oral	100.5 (92.5–114.6)	94.0 (90.5–97.5)	**0.002**	0.631
	ProMoFasT	109.5 (91.0–112.1)	97.0 (84.0–102.0)	**0.001**	
HbA1c	Oral	5.8 (5.5–6.1)	5.4 (5.3–5.7)	**0.005**	**0.009**
	ProMoFasT	6.2 (5.7–6.7)	6.2 (5.7–6.4)	**0.015**	
HOMA-IR	Oral	6.2 (4.7–9)	3.6 (1.5–7.4)	**0.027**	0.364
	ProMoFasT	5.0 (2.9–6.6)	2.8 (1.5–4.1)	**0.005**	
Fasting blood insulin	Oral	26.4 (14.6–32.1)	28.0 (22.0–30.2)	**0.028**	**0.001**
	ProMoFasT	19.4 (9.8–25.0)	11.8 (8.0–18.0)	**0.002**	
Creatinine	Oral	0.8 (0.7–1.0)	0.8 (0.6–1.1)	0.218	0.631
	ProMoFasT	0.8 (0.6–0.8)	0.8 (0.7–0.8)	0.550	
GGT	Oral	29.1 (21.6–51.2)	26.0 (18.0–36.0)	0.444	0.286
	ProMoFasT	19.2 (16.0–35.4)	19.0 (17.0–29.5)	0.513	
ALP	Oral	124.6 (67.0–190.9)	128.0 (78.0–199.0)	**0.018**	0.569
	ProMoFasT	100.5 (66.0–183.8)	101.5 (84.0–168.0	**0.016**	
AST	Oral	20.1 (15.6–26.6)	21.0 (16.6–40.0)	0.153	0.201
	ProMoFasT	17.1 (16.0–22.1)	17.0 (13.5–20.0)	0.055	
ALT	Oral	29.1 (17.1–41.7)	20.0 (14.0–35.0)	**0.016**	0.841
	ProMoFasT	19.6 (17.6–46.7)	18.5 (17.0–29.0)	**0.011**	
TC	Oral	194.0 (177.3–238.6)	218.1 (184.0–237.5)	**0.004**	0.314
	ProMoFasT	187.9 (171.7–241.0)	181.4 (159.9–229.5)	**0.001**	
HDL–C	Oral	53.3 (47.5–56.0)	55.9 (54.6–57.8)	**0.005**	0.722
	ProMoFasT	49.2 (44.4–57.0	53.0 (46.7–60.4)	**0.002**	
LDL–C	Oral	129.6 (99.5–167.8)	143.7 (103.8–149.1)	**0.012**	0.539
	ProMoFasT	116.6 (94.9–158.0)	107.7 (93.8–161.4)	**0.099**	
TG	Oral	127.1 (98.5–192.4)	98.5 (67.0–165.0)	**0.005**	0.821
	ProMoFasT	119.6 (75.8–200.0)	94.5 (52.3–151.0)	**0.002**	
Calcium	Oral	9.0 (8.9–9.7)	9.3 89.2–10.0)	**0.005**	0.310
	ProMoFasT	9.4 (9.2–9.6)	9.7 (9.4–9.9)	**<0.0001**	
Phosphorus	Oral	3.1 (2.9–3.3)	3.2 (3.1–3.4)	**0.005**	0.643
	ProMoFasT	3.4 (2.9–4.4)	3.5 (2.6–4.2)	**0.001**	

Data are expressed as median and interquartile range. No significant differences were found between the two groups at baseline. Abbreviations: ProMoFasT: Protein-Sparing Modified Fast diet by nasogastric tube; HbA1c: glycated hemoglobin; HOMA-IR: homeostasis model assessment of insulin resistance; GGT: gamma-glutamyl transferase; ALP: alkaline phosphatase; AST: aspartate amino transferase; ALT: alanine amino transferase; TC: total cholesterol; HDL-C: high-density lipoprotein cholesterol; LDL-C: low-density lipoprotein cholesterol; TG: triglycerides. Bold font indicates statistical significance.

## Data Availability

Data for the reported results are available upon a reasonable request, in accordance with ethical principles.

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
