# Peer review of "The Real-Life Use of a Protein-Sparing Modified Fast Diet by Nasogastric Tube (ProMoFasT) in Adults with Obesity: An Open-Label Randomized Controlled Trial"

_nutrients, 2023, doi:10.3390/nu15224822_

Round 1

Reviewer 1 Report

Comments and Suggestions for Authors

An interesting and well planned study with relevant findings.

Below are some comments and suggestions:

-It would be helpful to share quality of life data if any were collected. This is especially important for the group with NG tubes

-All tables and figures should have acronyms spelled out at the bottom

-Figures should have p values added to explain the (*) symbols

-Line 53: replace commas with periods (1,2 - 1,5 should be 1.2-1.5)

-Table 2: explain numbers between brackets [e.g. numbers are mean (SD)]

-Discussion lines 287-289 make it sound that difference between groups was significant which is not what is reported in table 4

Author Response

An interesting and well planned study with relevant findings.

We thank the Reviewer for the positive evaluation of our work.

Below are some comments and suggestions:

-It would be helpful to share quality of life data if any were collected. This is especially important for the group with NG tubes.

Unfortunately, we did not collect quality of life data. Therefore, we implemented the limits section (Lines: 344-348), to point out that the lack of quality of life data could be a possible limitation of the study.

-All tables and figures should have acronyms spelled out at the bottom

We added abbreviations to the bottom of each table and figure, as suggested.

-Figures should have p values added to explain the (*) symbols

We thank the Reviewer for the observation and we change the (*) symbols with the p-value in each figure.

-Line 53: replace commas with periods (1,2 - 1,5 should be 1.2-1.5)

We replaced commas with periods as suggested

-Table 2: explain numbers between brackets [e.g. numbers are mean (SD)]

We clarified that the data are presented as a median and interquartile range by including a note at the bottom of each table.

-Discussion lines 287-289 make it sound that difference between groups was significant which is not what is reported in table 4

We thank the reviewer for the observation. We rephrased the sentence to better explain the results obtained.

Reviewer 2 Report

Comments and Suggestions for Authors

This study investigated the effect of a low carbohydrate, low fat ketogenic diet administered continuously via a nasogastric tube or taken orally. The conclusion is that the nasogastric diet resulted in a greater improvement in body composition and in metabolic markers. There were no differences in body weight, BMI or waist circumference between the two groups. Given that the diets were the same but administered differently is there an explanation as to why there was a dramatic reduction in fasting insulin in the nasogastric infusion group and no change or a slight rise in insulin in the oral groups.  

The two groups had different number of men,  25 % in the intragastric group and 41% in the oral group. This may have impacted on the muscle mass data. The same data is presented in the figures and table. Is it necessary to present the data both ways?

Minor points

Line 32.                                   millions should be million

Line 57                                    This statement implies that amino acids reduce insulin secretion the opposite is the truth. Insulin stimulates protein synthesis

Lnes 70 and 75  

and 83, 86                               Use people first.  “Obese patients” should be “patients with                        obesity”

Line 380.         “…with  sepsis of trauma.”” Should be “..with  sepsis OR Trauma

Line 183 Table 3          The ……. Should also contain the meaning of the parameters (e.g the meaning of BCM, FFM etc,

Line 223                      “Giving” should be “Given”

Line 308                      “For” should be “To”

Line 315                      “safety” should be “safe”

Comments on the Quality of English Language

Some minor English edits are required as suggested in the comments to the authors

Author Response

This study investigated the effect of a low carbohydrate, low fat ketogenic diet administered continuously via a nasogastric tube or taken orally. The conclusion is that the nasogastric diet resulted in a greater improvement in body composition and in metabolic markers. There were no differences in body weight, BMI or waist circumference between the two groups. Given that the diets were the same but administered differently is there an explanation as to why there was a dramatic reduction in fasting insulin in the nasogastric infusion group and no change or a slight rise in insulin in the oral groups.  

We thank the reviewer for the comment on our work.

The two groups had different number of men, 25 % in the intragastric group and 41% in the oral group. This may have impacted on the muscle mass data.

We thank the reviewer for the observation. We agree with him/her that the different number of men may slightly impact body composition assessment; thus, we added a sentence in the limit section about. However, in our sample, no significant differences were found between the two groups at baseline for all BIA parameters analyzed. We also added this last sentence at the bottom of the tables.

The same data is presented in the figures and table. Is it necessary to present the data both ways?

We thank the Reviewer for the constructive feedback. We believe presenting our data in both tabular and graphical forms serves distinct purposes and enhances the clarity and accessibility of our results. Including both allows readers to choose their preferred method for examining the data and ensures that the information is accessible to a broader audience with varying preferences for data consumption. We have ensured that the data in both presentations are complementary and add value to the understanding of the results.

Minor points

Line 32. millions should be million

We correct the term, as suggested.

Line 57. This statement implies that amino acids reduce insulin secretion the opposite is the truth. Insulin stimulates protein synthesis.

We thank the reviewer for the comment, and we agree with him/her. Accordingly, we rephrased the sentence to provide a clearer elucidation of Blackburn et al. rationale regarding the Protein-Sparing Modified Fast diet.

Lines 70 and 75  and 83, 86. Use people first. “Obese patients” should be “patients with obesity”

We correct the terms, as suggested. We also changed the title using people first.

Line 380.         “…with  sepsis of trauma.”” Should be “..with  sepsis OR Trauma

We thank the reviewer for the observation and we corrected the reference as follow: Blackburn, G.L.; Flatt, J.P.; Clowes, G.H.; O’Donnell, T.F.; Hensle, T.E. Protein Sparing Therapy during Periods of Starvation with Sepsis or Trauma. Ann Surg 1973, 177, 588–594.

Line 183 Table 3          The ……. Should also contain the meaning of the parameters (e.g the meaning of BCM, FFM etc,

We added abbreviations to the bottom of each table and figure, as suggested.

Line 223                      “Giving” should be “Given” 

Line 308                      “For” should be “To” 

Line 315                      “safety” should be “safe” 

We correct the terms, as suggested.

Some minor English edits are required as suggested in the comments to the authors.

We submitted the manuscript to an English language expert from the University of Genoa for the Minor editing of English required by the Reviewer.
